FDAAA legislation is working, but methodological flaws undermine the reliability of clinical trials: a cross-sectional study

Marin dos Santos Douglas H. douglas.santos@agu.gov.br
Atallah Álvaro N.
Federal University of São Paulo (Unifesp) , São Paulo , Brazil
Bentzen Søren
Electronic publication date: 2015 Jun 25
Publication date: 2015
Volume: 3
Electronic Location ID: e1015
Received 2015 Feb 25; Accepted 2015 May 21
Copyright: © 2015 Marin dos Santos and Atallah
Copyright year: 2015
Copyright holder: Marin dos Santos and Atallah
License: This is an open access article distributed under the terms of the Creative Commons Attribution License, which permits unrestricted use, distribution, and reproduction in any medium, provided the original author and source are credited.
License URL: https://creativecommons.org/licenses/by/3.0/

Keywords: FDAAA, Selective publication, Clinical trials, Design methods, Right to health, Right to information

Funding: The authors declare there was no external funding for this work.

==============================
The relationship between clinical research and the pharmaceutical industry has placed clinical trials in jeopardy. According to the medical literature, more than 70% of clinical trials are industry-funded. Many of these trials remain unpublished or have methodological flaws that distort their results. In 2007, it was signed into law the Food and Drug Administration Amendments Act (FDAAA), aiming to provide publicly access to a broad range of biomedical information to be made available on the platform ClinicalTrials (available at https://www.clinicaltrials.gov). We accessed ClinicalTrials.gov and evaluated the compliance of researchers and sponsors with the FDAAA. Our sample comprised 243 protocols of clinical trials of biological monoclonal antibodies (mAb) adalimumab, bevacizumab, infliximab, rituximab, and trastuzumab. We demonstrate that the new legislation has positively affected transparency patterns in clinical research, through a significant increase in publication and online reporting rates after the enactment of the law. Poorly designed trials, however, remain a challenge to be overcome, due to a high prevalence of methodological flaws. These flaws affect the quality of clinical information available, breaching ethical duties of sponsors and researchers, as well as the human right to health.

Background and Introduction

Medical treatment, as a general rule, must rely on the best available clinical evidence. The strength of a therapeutic recommendation is a complex process, but usually systematic reviews of high quality randomized controlled trials are accepted as the gold standard (Gülmezoglu & Villar, 2003).

Broad and free access to biomedical research is therefore essential to public health and individual clinical decisions. However, many researchers and sponsors of clinical trials have been negligent in disclosing their findings, impairing transparency of biomedical research.

Lack of transparency in clinical research has many different faces. However, it usually emerges in two respects. First, researchers and sponsors hide results from the public by failing to report or publish their findings or by publishing partial or fraudulent scientific papers. These behaviors are usually called selective publication of clinical trials.1

Second, many clinical trials have methodological flaws that stop them from being a fair and ethical test on whether the tested therapy is working (Goldcare, 2014). Poor methodological designs allow results to be manipulated in ways that distort benefits and risks, according to the purposes of sponsors or researchers. They will provide neither reliable results nor valid conclusions (Jadad et al., 1996). Such biased trials are a mere sham of evidence, unable to determine whether the assessed intervention is effective, efficient, or safe.

Selective publication and poorly designed trials can lead to tragic outcomes. Doctors and patients are misled, and policymakers are misinformed, resulting in ungrounded clinical and policy decisions. Risks of new drugs may be underestimated, efficacy may be overestimated, and the risk–benefit ratio can be changed, resulting in potentially life-threatening decisions and disastrous policy options (Turner et al., 2008). Such practices may endanger the right to health and undermine evidence-based medicine, breaching the ethical duties of researchers and sponsors (Gøtzsche, 2012; Every-Palmer & Howick, 2014).

Lack of transparency affects a large proportion of clinical trials, both ongoing and completed, especially industry-funded trials (Bekelman, 2003). According to Gøtzsche (2012), 9 out of the 10 largest pharmaceutical companies were sued and signed corporate integrity agreements under civil and criminal law, due to unethical and unlawful practices in the United States. Between 1987 and 2010, the American government recovered more than US$18 billion as a result of frauds associated with healthcare cases (US Dep. of Justice, 2010). Considering the US budget for healthcare (Carter & Cox, 2011) authorities believe that up to 8% is lost due to fraudulent practices (Grassley, 2011).

Abuses in biomedical research, however, cannot persist due to the potential harm to research participants, patients, and the population as a whole. The nexus between transparency, information, and the right to health is clear from the individual’s perspective and is within the scope of public health. Information enables individuals to promote their own health, claim for quality services and adequate policies, control and follow the progressive realization of their rights, and consent freely about their own bodies and health (Neto, 2004). Adequate information also enables policymakers to build evidence-based guidelines, managing public health and its scarce resources on an optimal scale.

It means that any patient, health professional, researcher, and policymaker, individually or organized in groups, have the right to access information on available medicines and therapies, including the effectiveness, side effects, and risks. Researchers, sponsors, and governments, therefore, have the power and duty to comprehensively and accurately make health information publicly available (MacNaughton & Hunt, 2006; Lemmens & Telfer, 2012; World Health Organization, 2015).

Many efforts have been made to accomplish broader trial transparency. In 1997, the Food and Drug Administration Modernization Act (FDAMA, Section 113) launched the ClinicalTrials.gov website and required the registration of protocols of clinical trials in order to disclose the study’s objectives, design, methods, relevant scientific background, and statistical information (ClinicalTrials.gov, 2014). ClinicalTrials.gov is maintained by the National Library of Medicine (NLM) at the National Institutes of Health (NIH) and “provides patients, their family members, health care professionals, researchers, and the public with easy access to information on publicly and privately supported clinical studies on a wide range of diseases and conditions” (ClinicalTrials.gov, 2013).

In 2004, the International Committee of Medical Journal Editors (ICMJE) required trial registration on ClinicalTrials.gov in order to consider manuscripts for publication in any of its member journals. In 2008, the revised Declaration of Helsinki (DoH) stated that every clinical trial must be registered before recruitment of the first subject. These attempts—among others—were not enforced by penalties. Thus, although registration was enhanced, it was not as broad and comprehensive as policymakers and authorities expected.

In 2007, the US Congress enacted the FDA Amendments Act (FDAAA) in order to expand the clinical trial registry database created by FDAMA. The FDAAA requires the registration of every protocol of clinical trials, other than phase 1, of any drug, biologic, or device that meets the legal definition of an applicable clinical trial.2 FDAAA Section 801 also requires mandatory reporting of clinical trial results on website ClinicalTrials.gov (http://clinicaltrials.gov) not later than 1 year after the primary completion date, defined as when “the last participant in a clinical study was examined or received an intervention and that data for the primary outcome measure were collected” (ClinicalTrials.gov, 2014). The penalty for the responsible party who fail to comply is up to US$10,000/day. According to the Title VIII of FDAAA 801, the responsible party is the “sponsor, sponsor-investigator, or sponsor-designated principal investigator who is responsible for submitting information about a clinical study to ClinicalTrials.gov and updating that information” (ClinicalTrials.gov, 2014). Both registration and results reporting must be achieved through the Protocol Registration System (PRS) of ClinicalTrials.gov (http://prsinfo.clinicaltrials.gov) (ClinicalTrials.gov, 2015).

Although the FDAAA was the first legislation enforced by monetary penalties, the literature claims it has not been effective in reaching broader public access to clinical information (Law, Kawasumi & Morgan, 2011; Kuehn, 2012; Gill, 2012). Prayle, Hurley & Smyth (2012), in a study similar to ours, found that only 22% (163/738) of clinical trials registered on ClinicalTrials.gov reported results within the legal time frame. Nevertheless, the FDA did not acknowledge Prayle, Hurley & Smyth’s findings, suggesting that methodological flaws have biased the reliability of their research (Hawkes, 2012).

Thus, little is known regarding the methodological quality of clinical trials registered at ClinicalTrial.gov, the effectiveness of FDAAA 801, and its correlation with selective publication of clinical trials. Therefore, in order to contribute to the current literature, we decided to evaluate the patterns of transparency and the methodological quality of clinical trials registered on ClinicalTrials.gov. Due to its growing economic and therapeutic importance, we decided to assess only protocols of clinical trials of biological medical products. 3,4,5 In particular, we evaluated the top five global best selling monoclonal antibodies (mAb) adalimumab, bevacizumab, rituximab, trastuzumab, and infliximab.

We also examined the methodological quality of these studies in order to evaluate whether industry-funded trials have a poorer design, biasing the findings. Additionally, we evaluated the impact of FDAAA 801 on researchers’ decisions to report results on ClinicalTrials.gov and publish their findings in scientific medical journals.

Methods

We performed an analytical cross-sectional study with data collection on the ClinicalTrials.gov website and on the databases PubMed, Embase, Lilacs, Cochrane Central, and Google Scholar.

On ClinicalTrials.gov, we searched for registered protocols of clinical trials on adalimumab, bevacizumab, rituximab, trastuzumab, and infliximab (see our search strategy and exclusion criteria in Appendix S1).

Based on the data gathered on ClinicalTrials.gov, in order to find whether completed trials were published, we searched for corresponding papers in journals indexed by PubMed, Embase, Lilacs, Cochrane Central, and Google Scholar (see our search strategy in Appendix S2).

We then extracted the relevant data from both protocols and published papers.

Outcomes

In order to establish the current patterns of selective publication of clinical trials, we assessed the proportion of published and unpublished trials and then associated our findings to the type of funding for each study. We also evaluated the proportion of positive, negative, partially positive, neutral, or inconclusive results among published papers.

In addition, we assessed the proportion of trials that have reported results on ClinicalTrials.gov and then associated our findings with the type of funding for each protocol.

In order to establish the methodological quality of the protocols, we assessed the proportion of single arm studies, (defined as a trial in which all participants receive the same intervention (ClinicalTrials.gov, 2014)), placebo-controlled studies (defined as a trial in which a group of participants receives a placebo (ClinicalTrials.gov, 2014)), and usual therapy-controlled studies (defined as a trial in which a group of participants receives a comparison drug that is considered to be effective (ClinicalTrials.gov, 2014)). We then correlated these findings with type of funding of each protocol (industry or independently funded).

Furthermore, we evaluated the type of masking used (if any), the randomization of participants (if any), and the control groups (if any). At this point, we used the Jadad scale (Jadad et al., 1996) to evaluate the methodological quality of the clinical trials in our sample. We acknowledge the Jadad scale has some limitations,6 and we don’t ignore there are several other scales and checklists for quality assessment, such as the Delphi list (Verhagen et al., 1998), the CONSORT 2010 statement (Schulz, Altman & Moher, 2010), and the Cochrane Collaboration’s tool (Higgins et al., 2011). Nevertheless, we chose the Jadad scale because it is reliable, validated, and easy to use and understand, even for those who do not have specific training on clinical trials assessment (Sjögren & Halling, 2002; Olivo et al., 2008).

Finally, so as to determine the effectiveness of FDAAA 801, we divided the original sample (n = 243) into three different subgroups: (a) studies completed before the enactment of FDAAA 801, (b) studies completed after FDAAA 801, but not covered by mandatory reporting, and (c) studies completed after FDAAA 801 and covered by mandatory reporting. We then assessed the proportion of published trials and reported results in each subgroup.

Results

According to the search strategies described in Appendix S1, we found 442 protocols of clinical trials registered on ClinicalTrials.gov of the biologics adalimumab, bevacizumab, rituximab, trastuzumab, and infliximab. We excluded 199 protocols according to our exclusion criteria7 (see also Appendix S1). There were 243 protocols remaining: adalimumab (n = 67), infliximab (n = 65), rituximab (n = 54), bevacizumab (n = 43), and trastuzumab (n = 14).

Publication of clinical trials, reporting of clinical trial results on ClinicalTrials.gov, and funding sources of clinical trials

Regarding our sample of 243 protocols, we compared the proportion of published and unpublished studies. Through December 31, 2013, 178 clinical trials were published (≈73.3%) while 65 remained unpublished (≈26.7%). The proportion of published papers for each biologic were as follows: adalimumab (n = 54/67; ≈ 80.6%), infliximab (n = 50/65; ≈ 77%), rituximab (n = 36/54; ≈ 66.6%), bevacizumab (n = 29/43; ≈ 67.4%) and trastuzumab (n = 10/14; ≈ 71.4%).

With regard to reporting results on ClinicalTrials.gov, we found that only 73 trials (≈30%) reported results online, as required by FDAAA 801. By cross-referencing our findings, we also identified 38 trials (≈15.6%) that were neither published nor reported at ClinicalTrials.gov. In that situation, data on clinical trials is entirely absent, in a way that it is not possible to assert whether the tested biologic works properly, if it is cost-effective, safe, and adequate for the condition of the patient.

Regarding the type of funding, we found 169 (≈70%) industry-funded trials and 74 (≈30%) independently funded trials (studies not funded by the pharmaceutical industry). Among unpublished clinical trials (n = 65), 44 (≈67.7%) were industry-funded while 21 (≈32.3%) were independently funded.

Does FDAAA 801 work? Possible impacts of US legislation on subgroups S1, S2, and S3

According to FDAAA 801, reporting results on ClinicalTrials.gov is mandatory up to 12 months after the completion of the study. Regarding our sample of 243 studies, only 73 (≈30%) reported results online. Nevertheless, FDAAA 801 does not cover all trials registered on ClinicalTrials.gov, but only applicable clinical trials.8

Therefore, in order to find whether FDAAA 801 has positively affected publication and reporting rates of the clinical trials assessed, we divided our original sample into three different subgroups. The first subgroup (S1) comprised trials completed before the FDAAA (2002 through 2006). The second subgroup (S2) comprised trials completed after the FDAAA (2008 through 2012), but not covered by mandatory reporting, and the third subgroup (S3) comprised trials completed after the FDAAA (2008 through 2012) and under mandatory reporting (Fig. 1).

Figure 1 Definition of subgroups S1, S2, and S3.

Regarding this specific outcome, we excluded all trials completed prior to 2002 and after 2012. We also excluded trials for which the completion date or, alternatively, the primary completion date, was not available on ClinicalTrials.gov (n = 55).

In subgroup 1 (n = 44), 28 trials (≈63.6%) were published and 6 trials (≈13.6%) reported results on ClinicalTrials.gov. In subgroup 2 (n = 87), 61 trials (≈70.1%) were published and 31 trials (≈35.6%) reported results on ClinicalTrials.gov. Finally, in subgroup 3 (n = 57), 48 trials (≈84.2%) were published and 40 trials (≈70.2%) reported results on ClinicalTrials.gov.

When we compared subgroup 1 (trials completed prior to FDAAA enactment) with subgroup 3 (trials under mandatory reporting), the proportion of published studies significantly increased from ≈63.6% to ≈84.2% (p = 0.032) and the proportion of reported results rose from ≈13.6% to ≈70.2% (p < 0.001) (Tables 1 and 2).

Table 1 Proportion of reported and unreported results on ClinicalTrials.gov (subgroups S1, S2, and S3, ≈%).

Subgroup	Unreported	Reported	Total	
	n	%	n	%	n	%	
S1	38	86.4	6	13.6	44	100	
S2	56	64.4	31	35.6	87	100	
S3	17	29.8	40	70.2	57	100	

Table 2 Proportion of published and unpublished trials (subgroups S1, S2, and S3, ≈%).

Subgroup	Unpublished	Published	Total	
	n	%	n	%	n	%	
S1	16	36.4	28	63.6	44	100	
S2	26	29.9	61	70.1	87	100	
S3	9	15.8	48	84.2	57	100	

On the other hand, when we compared subgroup 2 (trials not under mandatory reporting) with subgroup 3 (trials under mandatory reporting), the proportion of published papers (≈70.1% versus ≈84.2%, p = 0.084) and reported results (≈35.6% versus ≈70.2%, p < 0.001) was also increased (see Tables 1 and 2).

These findings suggest that FDAAA 801 may be positively influencing the proportion of published trials and reported results.

We also assessed the proportion of studies that were (a) both published and reported, (b) only published, (c) only reported, and (d) neither published nor reported. Our findings corroborate the above conclusion towards the effectiveness of FDAAA 801, as shown below in Table 3.

Table 3 Proportion of Clinical Trials: (a) both published and reported, (b) only published, (c) only reported, and (d) neither published nor reported (missing data) (≈%).

	Subgroup 1	Subgroup 2	Subgroup 3	
	Pre-FDAAA 801	Not under mandatory reporting	Under mandatory reporting	
	(n = 44)	(n = 87)	(n = 57)	
(a) Trials both published and reported	n = 1 (≈2.3%)	n = 17 (≈19.5%)	n = 34 (≈59.7%)	
(b) Studies published only	n = 27 (≈61.3%)	n = 44 (≈50.5%)	n = 14 (≈24.5%)	
(c) Results reported only	n = 5 (≈11.4%)	n = 14 (≈16.1%)	n = 6 (≈10.5%)	
(d) Missing data	n = 11 (25%)	n = 12 (≈13.8%)	n = 3 (≈5.2%)	
Total	n = 44 (100%)	n = 87 (100%)	n = 57 (100%)	

Positive and negative results among published trials

In order to find whether clinical trials with positive results are more likely to be published when compared with trials that have negative or neutral results as alleged by the literature (Rising et al., 2008), we screened our subsample of published trials (n = 178) and evaluated each paper and its conclusions. That said, positive results were found in 118 papers (≈66.3%), while negative results were described in 18 papers (≈10%). Neutral or inconclusive results were reported in 11 trials (≈7%), and partially positive findings were described in 24 trials (≈13.5%). In 7 trials (≈4%), the same biologic was tested using different dosages or different forms of administration.

Substances assigned to the control group and their relation to funding sources

According to the World Medical Association Declaration of Helsinki, the “benefits, risks, burdens and effectiveness of a new intervention must be tested against those of the best proven interventions.” It means that within the context of an appropriately designed clinical trial, the new drug must be compared with a competitor that is known to be effective and safe (hereafter treatment as usual or TAU), in order to demonstrate the advantages or disadvantages of the new intervention.9

It is common, however, to compare the new intervention with a useless placebo substance, potentially distorting and biasing the results of the trial. It is also common, in worse scenarios, to find single arm studies (SAS), in which every participant enrolled receives the same experimental therapy.

Thus, in order to find the proportion of single arm studies, placebo-controlled, and TAU-controlled trials in our sample (n = 243), we assessed each protocol to determine the type of substance assigned as a control.

We found 84 (≈35%) single arm trials, 53 (≈22%) placebo-controlled trials, and 80 (≈33%) TAU-controlled trials. We also found 13 (≈5%) trials in which the new intervention was compared with placebo and TAU, and 13 (≈5%) trials in which the intervention was tested using different dosages or administration forms.

We then cross-referenced these findings with the type of funding for each clinical trial (industry-funded or independently funded) in order to find whether the source of funding affects, in any form, the design and reliability of the study (Table 4).

Table 4 Substances assigned to the control group according to the funding sources of each trial (industry-funded or independently funded).

	TAU	Placebo	TAU and placebo	Single arm	Different dosages or administration forms	Total	
	(n = 80)	(n = 53)	(n = 13)	(n = 84)	(n = 13)	(n = 243)	
Industry-funded (n = 169)	n = 44 (≈26%)	n = 44 (≈26%)	n = 13 (≈7.7%)	n = 62 (≈36.7%)	n = 6 (≈3.5%)	n = 169 (100%)	
Independently funded (n = 74)	n = 36 (48.6%)	n = 9 (≈12.2%)	–	n = 22 (≈29.7)	n = 7 (≈9.5%)	n = 74 (100%)	

We found a higher prevalence of single arm studies (62/169; ≈36.7% versus 22/74; ≈29.7%, p = 0.367) and placebo-controlled trials (44/169; 26% versus 9/74; ≈12.2%, p = 0.025) among industry-funded trials. On the other hand, we found TAU-controlled trials are more prevalent within independently funded trials when compared to industry-funded trials (36/74; ≈48.6% versus 44/169; ≈26%, p < 0.001).

Methodological design and quality of protocols

Based on the information gathered on ClinicalTrials.gov, we evaluated the methodological quality of protocols. First, we assessed whether the trial was a single arm design (i.e., no comparison group) or group-designed (i.e., participants are allocated in different groups). Second, we examined whether the trial randomly allocated participants in groups (randomization). Finally, we examined whether the clinical trial was masked to treatment allocation (i.e., double-blinded or single-blinded).

Out of the 243 protocols found, 159 (≈65.4%) allocated participants into two or more control groups and 84 (≈34.6%) were single arm trials. In addition, 149 (≈61.3%) trials were randomized and 94 (≈38.7%) were not randomized. Finally, 84 (≈34.5%) trials were blinded while 159 (≈65.4%) were not blinded.

Cross-referencing these findings, we determined that only 82 trials (≈33.7%) were cumulatively group-designed, randomized, and masked, achieving a good or fair methodological design according to the Jadad scale (Jadad et al., 1996).

At this point, it is noteworthy that the monoclonal antibodies adalimumab, bevacizumab, rituximab, and trastuzumab received orphan drug designation for the treatment of some rare diseases, according to the Orphanet website (http://www.orpha.net). It is known that the quality of clinical trials of rare conditions may be impaired, with remarkable differences in design, blinding and randomization (Bell & Smith, 2014). However, within our sample of 243 studies, only 11 (≈4.5%) were associated with the rare diseases referred by Orphanet.

Discussion

Our findings suggest that selective publication of clinical trials persists, regardless of the type of funding or intervention assessed (in our research, biologics). Through December 31, 2013, about 25% of clinical trials were not published and 15% were not published and did not have results reported on ClinicalTrials.gov.

Online reporting of results on ClinicalTrials.gov also remains low, ranging between 7% (prior to the FDAAA) and 70.2% (trials covered by mandatory reporting). Nevertheless, we found a significant increase in publication and reporting rates after FDAAA 801. These findings suggest that the US legislation is effective, achieving several of its goals.

Whereas the methodological quality of clinical trials is highly related to the transparency of clinical research, affecting its reliability and subsequent medical choices and health policies, we also assessed the methodological standards of registered studies. Not surprisingly, we found that approximately 67% (161/243) were graded as poor according to the Jadad scale. This means that only around one third of the protocols registered on ClinicalTrials.gov had a reliable methodological design (fair or good) (Jadad et al., 1996).

We also found that industry-funded trials are more likely to be single arm designed or placebo-controlled when compared to independently funded trials. On the other hand, TAU-controlled trials were more common among independently funded trials.

These findings suggest that, despite the fact that industry has been reporting and publishing its trials in similar proportions to those of independent researchers, poor methodological choices may undermine the reliability of industry-funded trials.

Finally, we also assessed the prevalence of positive, negative, neutral, or inconclusive results among published trials (n = 178). Positive (≈66.3%) and partially positive (≈13.5%) results were more prevalent, which is compatible with the literature (Decullier, Lhéritier & Chapuis, 2005; Rising et al., 2008). These findings may result from the prevalence of studies with poor methodological quality in our sample. After all, 137 (≈56.4%) trials were single arm designed or placebo-controlled trials and only 82 trials (≈33.7%) were blinded and randomized. On the other hand, because our findings are solely quantitative, it is possible that the prevalence of positive results is associated with the true efficacy of the tested biologics.

Although our findings suggest that FDAAA 801 has had positive impacts on the dissemination and expansion of biomedical information and data, it is important to highlight that poorly designed trials remain as a major challenge for transparency. This is because it is not easy to determine whether a clinical trial has methodological flaws, particularly for nonprofessionals. A poorly designed trial can distort results in ways that drug benefits are overestimated and risks or harms are underestimated, unacceptably breaching ethical and moral duties of sponsors and researchers.

Indeed, a database that requires protocol registration and results submission, but does not separate the “wheat from the chaff,” can potentially mislead health professionals, patients, and policymakers.10 Therefore, legislation needs to go further in order to require researchers and sponsors to provide ClinicalTrials.gov with data on the quality of clinical trials.

Thus, beyond study registration and results submission, we believe researchers and sponsors should be legally required to self-rate their protocols, according to the Jadad scale or other assessment system, in order to inform patients, health professionals, and policymakers about the methodological quality of each trial made publicly available.

We assume future legislation must address the subject as a growing demand for human rights-based medicine in which health decisions are made in light of comprehensive information. Any legal initiative, however, is likely to become useless if a single core value is not universally shared. Otherwise, clinical research may become discriminatory, because the basic rights and duties of participants and researchers will be different according to where the study is performed. Discrepant legal systems may lead to human rights violations or unfair financial inducement (Terwindt, 2014).

A global agenda on transparency must be homogenous and standardized, enabling broad access to results, methodological quality, and funding sources of clinical trials. Providing reliable, comprehensive, and easy access to data on biomedical research, beyond safeguarding the human right to health and information, enables the expansion of systematic reviews and, as a consequence, evidence-based medical and health decisions.

Strengths and limitations of this study

To our knowledge, this study is the first to evaluate the patterns of transparency of the clinical trials of biologics. We assessed different standards and trends associated with transparency: publication rates, reporting results on ClinicalTrials.gov, methodological flaws, and the impact of FDAAA 801 on clinical research. We also compared industry-sponsored trials with independently funded trials. Our findings set forth that, beyond reporting and publication bias, poor methodological quality of clinical trials is a challenge that must be faced in the near future.

Moreover, the methodology applied in our study was enhanced by the extensive search strategy for published papers employed on PubMed, Embase, Lilacs, Cochrane Central, and Google Scholar.

Finally, our research is aligned with recent World Health Organization Statement on Public Disclosure of Clinical Trial results (2015). The WHO statement establishes that researchers must publicly report results in both of the following two modalities. First, “main findings of clinical trials are to be submitted for publication in a peer-reviewed journal within 12 months of study completion.” Second, “key outcomes are to be made publicly available within 12 months of study completion by posting to the results section of the primary clinical trial registry” (World Health Organization, 2015). The available literature on the subject, to our knowledge, has primarily faced the patterns of results reporting at ClinicalTrials.gov (Zarin et al., 2011; Law, Kawasumi & Morgan, 2011; Kuehn, 2012; Gill, 2012; Prayle, Hurley & Smyth, 2012), which makes our study the first—or one of the first—to assess both reporting modalities currently recommended by WHO.

Our research, however, also has some limitations. First, we only assessed the protocols and publications of five biologics. Thus, because of sample bias, our findings may not represent the transparency patterns of clinical research as a whole. Indeed, the effectiveness of the FDAAA legislation may be limited to clinical trials of biologics.

In addition, we did not contact investigators (or other responsible parties) in order to confirm non-publication of their studies. We decided not to do so because contact details are not regularly disclosed on ClinicalTrials.gov. Furthermore, the literature suggests that investigators rarely answer questions about the publication of their trials (Stern & Simes, 1997; Decullier, Lhéritier & Chapuis, 2005; Ross et al., 2009).

Moreover, our findings may be partially biased due to incomplete or contradictory information posted on ClinicalTrials.gov (Chan, 2008; Ross et al., 2009; Smyth et al., 2011). However, we completed individual forms for each registered protocol that were manually checked for contradictions, potentially reducing bias risks.

It is important to highlight that our search strategy, even though based on the most significant available databases (PubMed, Embase, Cochrane Central, Lilacs, and Google Scholar), did not include any manual search of printed journals.

It is noteworthy that we had no information about studies that applied for exemptions from mandatory reporting on ClinicalTrials.gov. We also note that clinical trials of biologics previously approved by the FDA, but under investigation for new indications, are required to post results up to 2 years after completion. However, we were not able to identify these studies due to unavailable data at ClinicalTrials.gov.

Finally, we did not evaluate any other policies that could have influenced the outcomes assessed in this study. Nevertheless, we note that the FDAAA stands alone as the only legislation establishing monetary penalties for responsible parties who fail to comply with registration or results submission requirements.

Comparison with the literature

Our findings are consistent with the literature. However, regarding the impacts of the FDAAA on reporting rates on ClinicalTrials.gov, it is noteworthy that Prayle, Hurley & Smyth (2012), who used a similar methodology to this study, found significantly different results. According to their findings, only 22% of the results were posted online, while we found a significantly higher proportion of 70.2%. The imbalance may be explained by the following reasons: (a) Prayle’s sample was significantly larger and not limited to biologics; and (b) according to the FDA, Prayle’s sample was biased, because they included protocols that did not meet the legal definition of an applicable clinical trial.

Conclusion

Patterns of selective publication of clinical trials of biologics do not differ from other major classes of medical products. Funding sources did not affect publishing and reporting rates, but industry-funded trials were more likely to have methodological flaws when compared to independently funded trials. Most of the trials were performed under poorly designed protocols, lowering the accuracy and biasing the risk-benefit analysis.

Reporting of results and publication rates of clinical trials of biologics were enhanced under FDAAA 801. Expanding similar legal regulations worldwide should be an indelible goal for the near future, establishing a new legal and policy framework for the right to health and information.

Supplemental Information

Appendix S1 Search strategy for protocols registered on ClinicalTrials.gov

Click here for additional data file.

Appendix S2 Search strategy for published papers

Click here for additional data file.

Additional Information and Declarations

Competing Interests

Author Contributions

1 Selective publication of clinical trials covers different behaviors. The literature suggests that between 30% and 45% of registered clinical trials are never published (Turner et al., 2008; Ross et al., 2009). Positive results are more likely to be published when compared to negative results (Decullier, Lhéritier & Chapuis, 2005; Rising et al., 2008). Selective publication is derived from many causes, including negligence of researchers, concerns about financial losses (Decullier, Lhéritier & Chapuis, 2005; Turner et al., 2008), lack of international legislation about transparency in clinical research, and the complexity of peer-reviewed publication itself.

2 According to the ClinicalTrials.gov website, registration “is required for trials that meet the FDAAA 801 definition of an applicable clinical trial and were either initiated after September 27, 2007, or initiated on or before that date and were still ongoing as of December 26, 2007. Trials that were ongoing as of September 27, 2007, and reached the Completion Date before December 26, 2007, are excluded. (…) Applicable Clinical Trials generally include interventional studies (with one or more arms) of FDA-regulated drugs, biological products, or devices that meet one of the following conditions: the trial has one or more sites in the United States; the trial is conducted under an FDA investigational new drug application or investigational device exemption; [and] the trial involves a drug, biologic, or device that is manufactured in the United States or its territories and is exported for research.” Available at https://clinicaltrials.gov/ct2/about-studies/glossary (accessed 21 May 2014).

3 Biological products differ from drugs that are chemically synthesized in ways that affect their cost, production, administration, and clinical efficacy (Morrow & Felcone, 2004). According to the FDA website, biologics “include a wide range of products such as vaccines, blood and blood components, allergenics, somatic cells, gene therapy, tissues, and recombinant therapeutic proteins. Biologics can be composed of sugars, proteins, or nucleic acids or complex combinations of these substances, or may be living entities such as cells and tissues. Biologics are isolated from a variety of natural sources—human, animal, or microorganism—and may be produced by biotechnology methods and other cutting-edge technologies. Gene-based and cellular biologics, for example, often are at the forefront of biomedical research, and may be used to treat a variety of medical conditions for which no other treatments are available. (…) In contrast to most drugs that are chemically synthesized and their structure is known, most biologics are complex mixtures that are not easily identified or characterized. Biological products, including those manufactured by biotechnology, tend to be heat sensitive and susceptible to microbial contamination. Therefore, it is necessary to use aseptic principles from initial manufacturing steps, which is also in contrast to most conventional drugs. Biological products often represent the cutting-edge of biomedical research and, in time, may offer the most effective means to treat a variety of medical illnesses and conditions that presently have no other treatments available.” Available at http://www.fda.gov/AboutFDA/CentersOffices/OfficeofMedicalProductsandTobacco/CBER/ucm133077.htm (accessed 23 April 2015).

4 By 2016, biologics will account for an estimated 21% share of the global pharmaceutical market. Monoclonal antibodies adalimumab, bevacizumab, rituximab, trastuzumab, and infliximab are among the top ten global best-selling medical products with 2016 estimated sales of US 37.8 billion dollars (Reis, Landim & Pieroni, 2011).

5 Monoclonal antibodies adalimumab, bevacizumab, rituximab, trastuzumab, and infliximab are used for the treatment of a wide range of diseases including breast cancer, pancreatic cancer, non small-cell lung cancer, metastatic colon or rectum cancer, non-Hodgkin’s lymphoma, rheumatoid arthritis, psoriatic arthritis, plaque psoriasis, ankylosing spondylitis, Crohn’s disease, and macular degeneration (Dutta, 2009).

6 An adequate and comprehensive evaluation of clinical trials could assess up to ten different dimensions of the study. The Jadad scale rates only three of these dimensions, while Delphi rates six, Cochrane rates five, and the CONSORT report guide rates nine. Jadad scale, from this point of view, is less comprehensive and more subject to bias when compared to other similar tools (Berger & Alperson, 2009).

7 We excluded every clinical trial in which the assessed biologic was neither the primary intervention nor the primary comparator/control under evaluation (see Appendix S1 for detailed search strategy and exclusion criteria).

8 Only trials that meet the definition of an applicable clinical trial are under the purview of FDAAA 801. See note 2.

9 The 6th Revision of the Declaration of Helsinki states that the “benefits, risks, burdens and effectiveness of a new intervention must be tested against those of the best proven intervention(s),” except in cases where no proven intervention exists or where, for compelling methodological reasons, the use of placebo is necessary to determine the efficacy or safety of an intervention and the patients who receive placebo will not be subject to additional risks of serious or irreversible harm. Nevertheless, in October 2008 the FDA removed references to the DoH, in reaction to the restrictions on the use of placebo-controlled trials. According to the FDA, the US government “continues to support the Declaration’s underlying principles. However, (…) the US Government does not fully support the 2000 version of the Declaration because it contains certain statements that may be inconsistent with US law and policy (e.g., concerning use of placebos in clinical trials)” (Regulations.gov, 2009). Thus, from the perspective of the FDA, the use of placebo within the assessed sample is not an ethical violation, while the DoH points in the exact opposite direction. While we acknowledge that this is a controversial subject, it is noteworthy we found a significant higher prevalence of placebo-controlled studies among industry-funded trials, potentially revealing a link between economic interests and the use of placebo.

10 Berger & Alperson (2009) stress that “in many cases, flawed or misleading evidence is worse than no evidence at all. This is because the state of ignorance resulting from a lack of evidence is recognized as a state of ignorance, whereas the state of ignorance resulting from misleading evidence is not so recognized. In addition, the existence of any clinical trials, misleading or not, effectively precludes the possibility of planning future trials to address the same questions as those addressed by the existing trials. For these reasons, misleading evidence in the form of flawed clinical trials is quite troublesome to public health.”

The authors declare there are no competing interests.

Douglas H. Marin dos Santos conceived and designed the experiments, performed the experiments, analyzed the data, contributed reagents/materials/analysis tools, wrote the paper, prepared figures and/or tables, reviewed drafts of the paper.

Álvaro N. Atallah conceived and designed the experiments, analyzed the data, reviewed drafts of the paper.

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
