# Peer review of "FDAAA legislation is working, but methodological flaws undermine the reliability of clinical trials: a cross-sectional study"

_PeerJ, doi:10.7717/peerj.1015_

## Round 0.1 · original submission · Minor Revisions

I concur with the reviewers that this manuscript is a potentially useful addition to the published literature on the reliability of clinical trials. The manuscript would benefit from careful consideration of the suggestions from both reviewers. In particular, I agree that the limitations of the Jadad scale should be acknowledged and critically discussed.

·

Basic reporting

No Comments

Experimental design

No Comments

Validity of the findings

No Comments

Additional comments

This paper, which examines the level of transparency of clinical trials involving several biologics, makes a publishable contribution to a fast-growing body of literature on transparency. Its demonstration that industry-funded trials involving the biologics in question were more likely to be poorly designed from a methodological point of view is particularly noteworthy.

That said, it is my recommendation that Peer J require a number of revisions before accepting the paper for publication.

First, the abstract of the paper makes specific reference to Article 12 of the International Covenant on Economic, Social and Cultural Rights (ICESCR). It was therefore very surprising to see that the only other place that mentions ICESCR is in the endnotes to the paper rather than the main text. The authors should either develop their discussion and analysis of the ICESRC or drop it entirely from the paper. If they decide to incorporate it into their analysis, the fact that some countries, including the United States, are not currently signatories to the Covenant must also be addressed, especially given the centrality of US law to the paper’s argument (the authors discuss the FDAAA 2007 at some length and use its passage to structure their inquiry). The authors' inattention to this is particularly noteworthy on p. 9 (lines 5-6) where they state that they “assume future legislation must address the subject…” It is true that in some jurisdictions (particularly South America) there is a massive amount of litigation around the “right to health”. However, in other jurisdictions like the US and even Canada (which, unlike the US, is a signatory to ISESCR) the right to health has not led to major changes in access to health care services, much less information. The claims for access to information are quite different, in law, in those jurisdictions. The authors, in other words, should not make an assumption to the contrary without further argumentation.

Second, but on a related note, the authors’ discussion of “rights” (apart from the ICESCR) is weak throughout the paper. Whether greater transparency of clinical trials should be seen as part and parcel of a human right to health is a contested point, which, to my knowledge has only been developed in one or two publications to date. The authors more or less assert it as a fact in this paper, which is potentially misleading. My worry, then, is that international law and human rights is not within the realm of expertise of the authors. While it is perhaps fair to note the issue of rights in passing, I would encourage the authors to shy away from making specific legal claims absent greater analysis. I am not sure if space constraints allow those legal claims to be properly developed, so again it may be worthwhile to cut them from the paper entirely.

Third, there are a number of sentences or terms that are either unclear or inappropriate in the paper that the authors should correct or clarify:

• Page 1, lines 21-22 – the authors use the term ‘selective publication of clinical trials’ to capture not only publishing partial results, but also “fraudulent scientific papers”. In the literature, these two practices are generally not equated. Selective publication is used to capture the publication of only certain results. Fraudulent papers are, in contrast, typically used with reference to results that are entirely fabricated or manipulated. Both constitute deception but I think it’s important to make this distinction clear, if only to be consistent with the existing literature.
• Page 2, lines 30-31 – the statement is not accurate or at least unclear; several studies (albeit, of single drugs) demonstrate selective publication; if the authors are referring to the lack of more systematic or multi-drug studies of selective publication that should be made more explicit.
• Page 2, lines 33-34 – the authors should explain why they have chosen the drugs they have to examine.
• Page 3, line 13 – the term ‘dismembered’ is odd; suggest ‘divided’ instead.
• Page 3, line 24 – the authors refer to ‘exclusion criteria’ without saying what they are. Suggest including in the main text.
• Page 8, lines 13-14 – the authors should briefly explain how well known or widely used the “Jadad scale” is and whether it has any limitations. Further, if any of the biologics in question received orphan drug designations (for treatment of rare disorders) it may be important to qualify the importance of this scale given that certain features of sound trial design (randomization, blinding) may be challenging to achieve in clinical trials of rare conditions. Do the trials in the sample relate to such rare conditions? There are analyses on this issue in the literature (see responses to Kesselheim AS, Myers JA, Avorn J. Characteristics of Clinical Trials to Support Approval of Orphan vs Nonorphan Drugs for Cancer. JAMA: The Journal of the American Medical Association. 2011 Jun 8;305(22):2320–6), which raise this very concern and the authors should at least acknowledge.
• Page 8, line 26 – it isn’t clear what the phrase ‘loss of transparency’ is referring to. I suggest that the authors reword along the lines of “the poor methodological choices may not be sufficiently transparent.” Note, also, that this claim doesn’t follow from the trials they analysed because the methodological choices were indeed transparent based on the sources they examined. It’s just that those methodological choices may not be apparent to readers. The authors develop this point on pp. 8-9, so it may make more sense to have the phrase I struggled with (i.e. loss of transparency” come after that analysis on the bottom of p. 8 and top of p. 9.

If the authors are able to address the above concerns, I believe it should be accepted for publication. The authors contribution, particularly in highlighting a link between methodological rigor and industry sponsorship, is important. The argument in the paper would simply benefit from greater clarity in certain places and, unless the authors are able to explain and analyse the legal rights issues in greater depth, I suggest that they remove those references from the paper. I hope this feedback is helpful.

Sincerely,

Matthew Herder JSM LLM
Assistant Professor, Faculties of Medicine and Law
Health Law Institute, Dalhousie University

·

Basic reporting

The authors should be more explicit at the outset of the article (abstract, earlier in the introduction) that the study focuses on biologics. Further, since the study focuses on biologics, it would be useful to provide a definition of what a biologic is (in comparison to a regular pharmaceutical product).

The authors state that “the literature claims [the FDAAA] has not been effective in reaching broader public access to clinical information” (page 2, lines 24-26) but then only lists a single study by Payle, Hurley & Smythe to support this claim. Please provide more sources to support the claim.

Minor comments on word choice:

Page 3, line 13: dismembered has a negative connotation, consider just saying “divided”

Page 10, line 2: dissemblance – do you mean imbalance?

The sentence “Missing data, in that situations, is complete” is confusing (page 3, line 38). Do you mean that data on clinical trials was entirely absent in this situation?

Experimental design

While the authors specify that the study focuses on five specific biologics, they provide little rationale as to why these specific five products were chosen, other than to stating in the endnote that these biologics “have a significant economic role in the pharmaceutical market.” Are these five biologics, for example, the most profitable biologics, or the most widely prescribed? Some explanation for why this particular cross-section of drugs was selected should be provided.

In the first paragraphs of the Results section, the authors state that 199 protocols were excluded according to their exclusion criteria. Any such exclusion criteria should be set out in the methods sections. The current explanation set out in endnote 15 is confusing. I was not sure if the authors meant to state that they excluded protocols where biologic in question was not the primary product under study. Consider rewording this part to clarify.

The authors should clarify what is specifically required under section 801 for reporting results online: do these results have to be specifically published on clinicaltrials.gov, or can they be made available on other online sources (e.g. independent web pages). For example, on page 4, line 20, it states that trials for subgroup 2 were reported on clinicaltrials.gov or “elsewhere on the Internet”. The specific online reporting requirements of the FDAAA should be made more explicit.

Validity of the findings

At the bottom of page 4 (line 22 onwards), the authors state that “These findings demonstrate that FDAAA 801 strongly influenced the proportion of published trials and reported trials.” This is perhaps an overbroad assertion. The study findings may suggest a strong correlation between the requirements of Section 801 and clinical trials reporting, but there are a range of other policies that have been introduced in recent years that may have also influenced the patterns of clinical trials registration, such as requirements introduced by various peer-reviewed journals that require published studies to be registered. Further, the present study only looks at 5 specific biologics, so the conclusions can only be drawn for this the registration of clinical trials for this particular subset of the pharmaceutical market. The authors need to more carefully circumscribe the conclusions that they draw from their study. While valuable conclusions can certainly be drawn from this study about the impact of section 801 on clinical trials registration, these findings must be more closely tied to the specific products studied.

On page 7, the authors state that “in an appropriately designed clinical trials, the new drug must be compared with a competitor that is known to be effective and safe.” While I generally agree with this statement, it is worth noting that there is still significant controversy on this point, to the extent that the FDA has decided not to recognize the more recent revisions to the Declaration of Helsinki that declare placebo-controlled trials to be unethical in many circumstances. Since the US legislation is the specific focus of this study, it would be worth mentioning this controversy.

On page 8, line 35-36, the authors state that “its methodological choices demonstrate a clear and dramatic loss of transparency.” While I fully agree that poor clinical trial methodology may significantly decrease the value of the resulting data (a fact that manufacturers may be able to make strategic use of), this is a broader issue that goes beyond transparency. (Transparency is useful for revealing methodological limitations in clinical trials, but additional measures beyond transparency requirements such as clinical trials registration and results reporting will likely be required to tackle this problem.) Indeed, the study results seem to suggest that poor methodology is not a problem that is likely to be effectively dealt with through clinical trials registration and reporting requirements, so an expansion on this point would be worthwhile.

Since the parameters of the Jadad scale are relied on by the authors in analyzing the quality of different protocol designs, this scale should be more comprehensively described in the study methods. It would also be worthwhile to provide some support on how widely recognized the Jadad scale is as a measure of clinical trial quality, particularly considering that the authors suggest later in the discussion that the researchers should self-report the quality of their studies on the Jadad scale in the clinical trial registry.

Additional comments

The article is clearly and concisely presented and is a valuable addition to the literature around clinical trials transparency. The authors provide a good background summary on the need for enhancing the transparency of clinical trials and the study undertaken directly contributes to the knowledge base on this issue.

---

## Round 0.2 · accepted · Accept

Thank you for addressing the reviewers' concerns.